# Modelling the sensitivity of ice loss to calving front retreat rates in the Amundsen Sea Embayment, West Antarctica

Jowan M. Barnes[1,2], G. Hilmar Gudmundsson[1], Daniel N. Goldberg[2], and Sainan Sun[1]

[1]School of Geography and Natural Sciences, Northumbria University, Newcastle upon Tyne, UK
[2]School of Geosciences, University of Edinburgh, Edinburgh, UK

**Correspondence:** Jowan M. Barnes (jowan.barnes@northumbria.ac.uk)

**Abstract.** Ice sheet modelling studies of the Amundsen Sea Embayment (ASE) in West Antarctica have provided estimates of its future impacts on sea level rise. However, many of these studies have not considered the impacts of calving, a key process in the dynamics of marine-terminating glaciers. Sensitivity to calving front retreat is not well understood, so we set out to investigate it in systematic manner. In this study, we quantify the sensitivity of modelled future mass loss to ice front retreat in the ASE, including Pine Island and Thwaites Glaciers. We find that prescribing constant frontal retreat rates from 0.1 to 1 km a$^{-1}$ progressively increases the contribution to sea level rise when compared to experiments with a fixed ice front. The result with our highest rate of retreat is up to 21.4 mm additional sea level contribution by 2100, and 239 mm by 2300. We identify specific buttressing thresholds where loss of contact with bedrock features causes changes in the ice dynamics. These are reached at different times depending on the retreat rate, and are the main cause of sensitivity to movement of the ice front. We compare the variability in our results using different retreat rates to that when using ocean melt and surface mass balance (SMB) forcing derived from different earth system models for ISMIP6, as these climate forcings are major factors in determining the future evolution of the Antarctic ice sheet. We find that the variability due to these two factors is similar. We also find that the additional loss of ice due to a prescribed retreat rate is not heavily dependent on mass balance forcing, so can be quantified independently of the ocean-induced melt and SMB. Our results demonstrate the importance of accurately representing calving processes in models, showing that they can be as important as climate forcing and therefore deserve a similar amount of attention in future model development work.

## 1 Introduction

Calving is a key process in the dynamics of the West Antarctic ice sheet, as it can change buttressing forces on the ice shelves that act as a control on the speed of upstream grounded ice sheets, which is potentially significant to the evolution of upstream ice flow (e.g. Depoorter et al., 2013; Pattyn et al., 2017). Over the last quarter of a century, the mass loss from Antarctica attributed to ice front retreat is almost the same as that attributed to ice-shelf thinning (Greene et al., 2022).

In ice-flow models, it is often easier and more practical to implement calving in long-term simulations via a continuous calving rate, rather than modelling individual calving events as they occur naturally. Several calving laws have been proposed from

which rates are calculated (or can be derived from calculated positions) depending on factors such as crevasse depth (Benn et al., 2007; Nick et al., 2010), strain rates (Levermann et al., 2012), divergence (Pollard et al., 2015), cliff height (Pollard et al., 2015; Crawford et al., 2021) and tensile stress (Morlighem et al., 2016). These are able to be parameterised for use in large-scale models, but do not necessarily have a solid physical basis and contain parameters which must be tuned for individual glaciers. Other approaches based on damage and fracture mechanics have been used in specific small-scale or analytical cases (Duddu et al., 2013; Krug et al., 2014; Yu et al., 2017), but are not formulated in a way which can be easily extended to general large-scale modelling (Choi et al., 2018). Several existing calving laws were compared in simulations of Greenland by Choi et al. (2018), and of Antarctica by Wilner et al. (2023), with no single law reproducing observed calving fronts consistently across all glaciers. Therefore, there is no consensus on a suitable calving law to use for predictive simulations, and many models continue to use a fixed ice front, or implement removal of floating ice below a prescribed minimum thickness. For example, in the recent model intercomparison project ISMIP6 (Seroussi et al., 2020), only three of the ten participating models implement a more complex calving scheme. One applies an approach based on strain rates (Levermann et al., 2012), while two other models use the calving law of Pollard et al. (2015).

Distinct from the question of calving laws is another related matter; the sensitivity of modelled glacier dynamics to calving. This is presumably a question of how much buttressing is lost by the removal of ice, and there are some recent examples of work in this area. Reese et al. (2018) investigated the response to instantaneous thinning of sections of Antarctic ice shelves. Higher responses to thinning were generally found closer to the grounding line. A similar observation is made by Morlighem et al. (2021), where sensitivity to perturbations in basal melt is seen to be higher near grounding lines and along the shear margins of Pine Island Ice Shelf. Mitcham (2022) systematically removed ice at different distances from grounding lines, finding that over 80% of the buttressing capacity of many glaciers is provided by the closest 15% of ice to the grounding line. In some previous studies, removal of all floating ice has been tested and shown to have a large impact on the future of ice sheets (e.g. Sun et al., 2020; Barnes and Gudmundsson, 2022).

Thwaites Glacier and Pine Island Glacier (PIG) in the Amundsen Sea Embayment (ASE), West Antarctica, are among the fastest evolving on the continent (Sutterley et al., 2014; Shepherd et al., 2018). Together they are contributing almost three quarters of the current ice loss from Antarctica (Rignot et al., 2019), and they hold enough ice between them to raise sea levels by over a metre (Rignot et al., 2002). The ASE also includes another pair of large ice shelves, Dotson and Crosson, which are fed by several smaller glaciers. The ice shelves in this region represent different configurations of floating ice, which makes the ASE an ideal area in which to investigate responses to calving front retreat. Pine Island Ice Shelf (PIIS) is contained within a bay and provides buttressing to upstream grounded ice, including a small ice stream we refer to as 'PIGlet' which enters PIIS from the west. The shelves of Dotson and Crosson are heavily buttressed by Bear Island, located downstream of the outlet glaciers. Thwaites ice shelf consists of a heavily damaged Western Ice Tongue and an Eastern Ice Shelf restrained by only a single pinning point, which may unpin entirely within the next decade (Wild et al., 2022). A labelled map of the region is

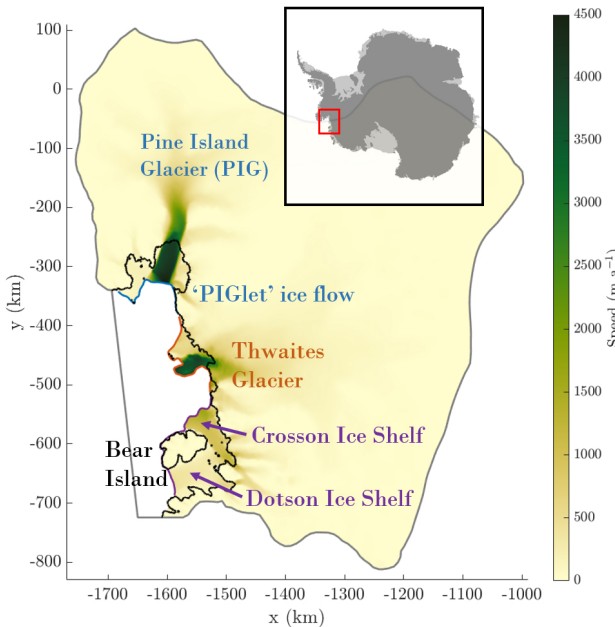

**Figure 1.** The Amundsen Sea Embayment labelled with its major features referred to in this work. The shading is the speed at the start of our simulations. The model domain boundary is indicated in grey, and the grounding line in black. The calving front is shown in colours matching the labels and corresponding to each catchment region in our discussion. The smaller insert shows the position of the region within Antarctica.

presented in Figure 1.

60

Our work explores the sensitivity of ice loss to calving front retreat by making use of recent advances in representation of ice front movement in a state-of-the-art ice-flow model. We investigate cases between the previously studied extremes of fixed calving fronts and instantaneous removal of all floating ice, in a way never previously done. Our key objective is to quantify the impact of frontal retreat rates on sea level contributions from ice loss, through systematic experiments.

65 ## 2  Experimental design

### 2.1  Model setup

We use the ice sheet model Úa (Gudmundsson, 2024), which implements the vertically-integrated Shallow Shelf Approximation (MacAyeal, 1989). The model solves simultaneously for the transient changes in ice thickness and ice velocities using a fully implicit time integration. Our domain covers the ASE region using inland boundaries based on the MEaSUREs Antarctic

Boundaries (Rignot et al., 2013), with smoothing applied. We use geometry from BedMachine Antarctica v3 (Morlighem et al., 2020), from which we derive the initial calving front. A shallow section of the bed downstream of the PIG grounding line is lowered to avoid an unrealistic initial advance and regrounding. Details can be found in Appendix A. Our bed geometry is shown in Figure 2(a). A Dirichlet condition is used to set all velocities along the inland boundary to zero, since the domain boundary generally follows the edges of drainage basins. The initial mesh is created using Mesh2D (Engwirda, 2014), with a resolution of 1 km at the grounding line, becoming coarser further upstream up to 10 km. An adaptive meshing scheme is applied such that the mesh is refined to 1 km around the grounding line and calving front as they move.

The densities are given the values of $917 \, \mathrm{kg\,m^{-3}}$ for ice and $1027 \, \mathrm{kg\,m^{-3}}$ for ocean water, consistent with the BedMachine dataset. The flow follows Glen's law (Glen, 1958) with exponent $n = 3$ and basal sliding follows a Weertman power law (Weertman, 1957):

$$\boldsymbol{\tau} = C^{-\frac{1}{m}} \|\boldsymbol{v}_b\|^{\frac{1}{m}} \frac{\boldsymbol{v}_b}{\|\boldsymbol{v}_b\|} \tag{1}$$

where $\boldsymbol{\tau}$ is the basal drag, $m = 3$ is a sliding exponent and $\boldsymbol{v}_b$ is the basal velocity. $C$ is a sliding parameter, which is inverted for along with the rate factor $A$ from Glen's law. We follow the inversion process detailed for Úa in Barnes et al. (2021).

The calving rate in Úa is defined as the difference between the retreat rate and the ice velocity normal to the calving front. The implementation of calving via a level-set method is presented in Appendix B.

## 2.2 ISMIP6 protocol

Following the ISMIP6-2300 experimental protocol (Seroussi et al., 2024), we begin our simulations in 2015 and run to 2300. Surface accumulation and basal melting, which we hereafter refer to together as climate forcing, are derived from ocean-atmosphere coupled simulations as specified in the protocol, using the local quadratic melting parameterisation with median MeanAnt calibration as set out in Jourdain et al. (2020). The equation for local quadratic melting, directly from the source, is

$$m(x,y) = \gamma_0 \times \left(\frac{\rho_{\mathrm{sw}} c_{\mathrm{pw}}}{\rho_{\mathrm{i}} L_f}\right)^2 \times \{\max[\mathrm{TF}(x,y,z_{\mathrm{draft}}) + \delta T_{\mathrm{sector}}, 0]\}^2, \tag{2}$$

where $\rho_{\mathrm{sw}}$ is sea water density, $\rho_{\mathrm{i}}$ is ice density, $c_{\mathrm{pw}} = 3974 \, \mathrm{J\,kg^{-1}\,K^{-1}}$ is the specific heat of sea water and $L_{\mathrm{f}} = 3.34 \times 10^5 \, \mathrm{J\,kg^{-1}}$ is the fusion latent heat of ice. $\mathrm{TF}(x,y,z_{\mathrm{draft}})$ is the thermal forcing provided by the ocean-atmosphere models, and relies on the ice draft in the ice model. The coefficient $\gamma_0$ and temperature correction $\delta T_{\mathrm{sector}}$ are used for calibration. Full details of the MeanAnt calibration which determines values for $\gamma_0$ and $\delta T_{\mathrm{sector}}$ are found in Jourdain et al. (2020), and summarised in Figure 3 within that paper. We do not reproduce the method here.

The seven 'Tier 1' experiments comprise the CMIP5 (Taylor et al., 2012) RCP8.5 scenario outputs from CCSM4 (Gent et al., 2011) and HadGEM (Collins et al., 2011), the CMIP5 RCP2.6 scenario output from NorESM (Iversen et al., 2013) with repeated forcing after 2100, the CMIP6 (Eyring et al., 2016) SSP5-8.5 scenario outputs from CESM2 (Danabasoglu et al.,

2020) and UKESM (Sellar et al., 2019), an additional UKESM output with repeated forcing after 2100 and a Control which applies constant thermal forcing throughout the simulation. The repeated forcing is taken from the 2080-2100 period, sampled randomly between 2100 and 2300 to avoid repeating the same forcing pattern.

## 2.3 Overview of experiments

Experiments are started with the present-day geometry of BedMachine. We prescribe constant, uniform retreat rates along the entire calving front of the ASE, ranging from 0 to $1\,\mathrm{km\,a^{-1}}$, in steps of $0.1\,\mathrm{km\,a^{-1}}$. The retreat rate of $0\,\mathrm{km\,a^{-1}}$ is specifically included as a control, in which the calving front does not move. Each retreat rate value is used in an experiment forced by the Control forcing, with further experiments for 0, 0.5 and $1\,\mathrm{km\,a^{-1}}$ run with each of the forcing scenarios. We refer to experiments by their climate forcing and retreat rate in the format 'Forcing_RR#', so for example the simulation using Control forcing with a retreat rate of $0.5\,\mathrm{km\,a^{-1}}$ would be Control_RR0.5.

We only allow calving on fully floating elements, since calving of grounded termini around Antarctica is minimal (Greene et al., 2022) and it would be unrealistic to apply the same retreat rate universally. This means that grounded ice is not removed, but any ice which comes afloat due to changing dynamics and geometry during the simulation is then subject to the prescribed retreat rate.

Additional experiments are run to identify whether behaviours can be attributed to particular parts of the calving front. This involves splitting the calving front into three sections as displayed in Figure 1; PIG, Thwaites and Crosson/Dotson. Experiments are then run using the Control forcing in which each of these sections is allowed to retreat individually, while the rest of the calving front remains in place.

An overview of all the simulations run is given in Table 1.

## 2.4 Sensitivity calculation

We calculate the sensitivity of negative changes in water-equivalent volume above flotation (VAF) - expressed as a contribution to mean sea level rise (SLR) - to prescribed retreat rates (RR), by quantifying an *SLR-RR sensitivity*, $\Upsilon$, defined as

$$\Upsilon = -\frac{\delta\mathrm{VAF}}{\delta\mathrm{RR} \times A_o}\,, \tag{3}$$

where $\delta\mathrm{VAF}$ and $\delta\mathrm{RR}$ are the differences between the VAF and RR values in two different experiments at the same model time, and $A_o = 3.614 \times 10^8\,\mathrm{km^2}$ is the global surface area of the ocean (Charette and Smith, 2010). The SI units of $\Upsilon$ are seconds, but to aid physical interpretation we express $\Upsilon$ in millimeters of sea level rise per metre of frontal retreat per year, i.e. as $\mathrm{mm\,(m\,a^{-1})^{-1}}$.

**Table 1.** An overview of all the experiments run for this work. RR represents the retreat rate in $\text{km a}^{-1}$. The climate forcings are those detailed in the text, with UKESMrep referring to the output using repeated forcing after 2100. Checkmarks indicate the combinations of parameters for which simulations were run. Double checkmarks are cases where additional simulations were run in which sections of the calving front were retreated individually.

|  | RR0 | RR0.1 | RR0.2 | RR0.3 | RR0.4 | RR0.5 | RR0.6 | RR0.7 | RR0.8 | RR0.9 | RR1 |
|---|---|---|---|---|---|---|---|---|---|---|---|
| Control | ✓✓ | ✓ | ✓ | ✓ | ✓ | ✓✓ | ✓ | ✓ | ✓ | ✓ | ✓✓ |
| CCSM | ✓ |  |  |  |  | ✓ |  |  |  |  | ✓ |
| CESM | ✓ |  |  |  |  | ✓ |  |  |  |  | ✓ |
| HadGEM | ✓ |  |  |  |  | ✓ |  |  |  |  | ✓ |
| NorESM | ✓ |  |  |  |  | ✓ |  |  |  |  | ✓ |
| UKESM | ✓ |  |  |  |  | ✓ |  |  |  |  | ✓ |
| UKESMrep | ✓ |  |  |  |  | ✓ |  |  |  |  | ✓ |

## 3 Results

### 3.1 Response of the ASE to calving front retreat

Figure 2(b-h) shows the difference in ice speed between Control_RR0.5 and Control_RR0 at various points in time during the simulation. These are presented alongside the bed geometry (Figure 2(a)) to help in interpreting aspects of the ice evolution, particularly with respect to where pinning points are located. This example demonstrates common features of the response of the ASE to calving front retreat across our ensemble. Generally the introduction of a calving front retreat rate leads to greater speeds and more loss of ice, but in some regions the ice becomes slower or thicker compared to Control_RR0. In the remainder of this section we summarise the responses of the three main regions within the domain.

Despite the central flow from the main trunk of PIG being faster when the retreat rate is higher, there is almost no grounding line movement here until all the floating ice downstream is removed (Figure 2(g)). However, 'PIGlet' speeds up in response to forced calving front retreat, and by 2100 the grounding line in Control_RR0.5 has already retreated in a way that does not occur in Control_RR0 during our simulation timeframe (Figure 2(e)). This grounding line retreat causes 'PIGlet' to merge with Eastern Thwaites, driving further retreat.

Thwaites ice shelf shows two types of response, with the eastern section being thicker and slower in Control_RR0.5, while the western section flows faster. As Eastern Thwaites merges with 'PIGlet' and undergoes grounding line retreat (Figure 2(e)),

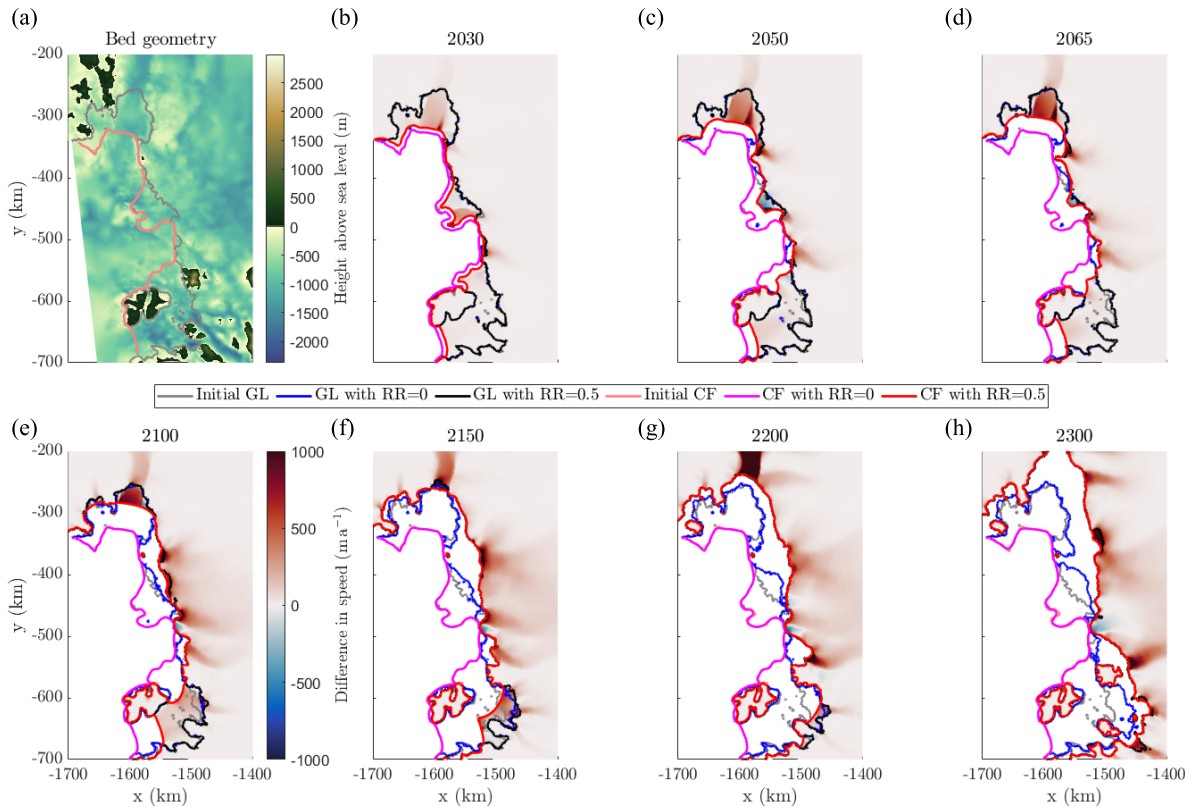

**Figure 2.** (a) The bed geometry from BedMachine. (b-h) Snapshots of differences in ice speed between Control_RR0.5 and Control_RR0 at different years of the simulation. Videos of the Control_RR0 experiment and the differences in ice speed and thickness above flotation for Control_RR0.5 (this figure) and Control_RR1 when compared to it are available as supplementary files.

the entire catchment ends up flowing faster. Western Thwaites does not undergo a significantly different grounding line migration compared to Control_RR0 until later in the simulation, after 2100 (Figure 2(f)).

150

The Crosson and Dotson ice shelves display very little reaction to a forced retreat rate initially, only starting to speed up significantly between 2100 and 2150 when the ice shelf loses contact with Bear Island (Figure 2(f)). Even then, the outlet glaciers do not show a large increase in speed compared to Control_RR0 until further towards the end of the simulation – after 2200 – as contact is lost with pinning points further upstream and the ice shelf is almost entirely removed (Figure 2(h)).

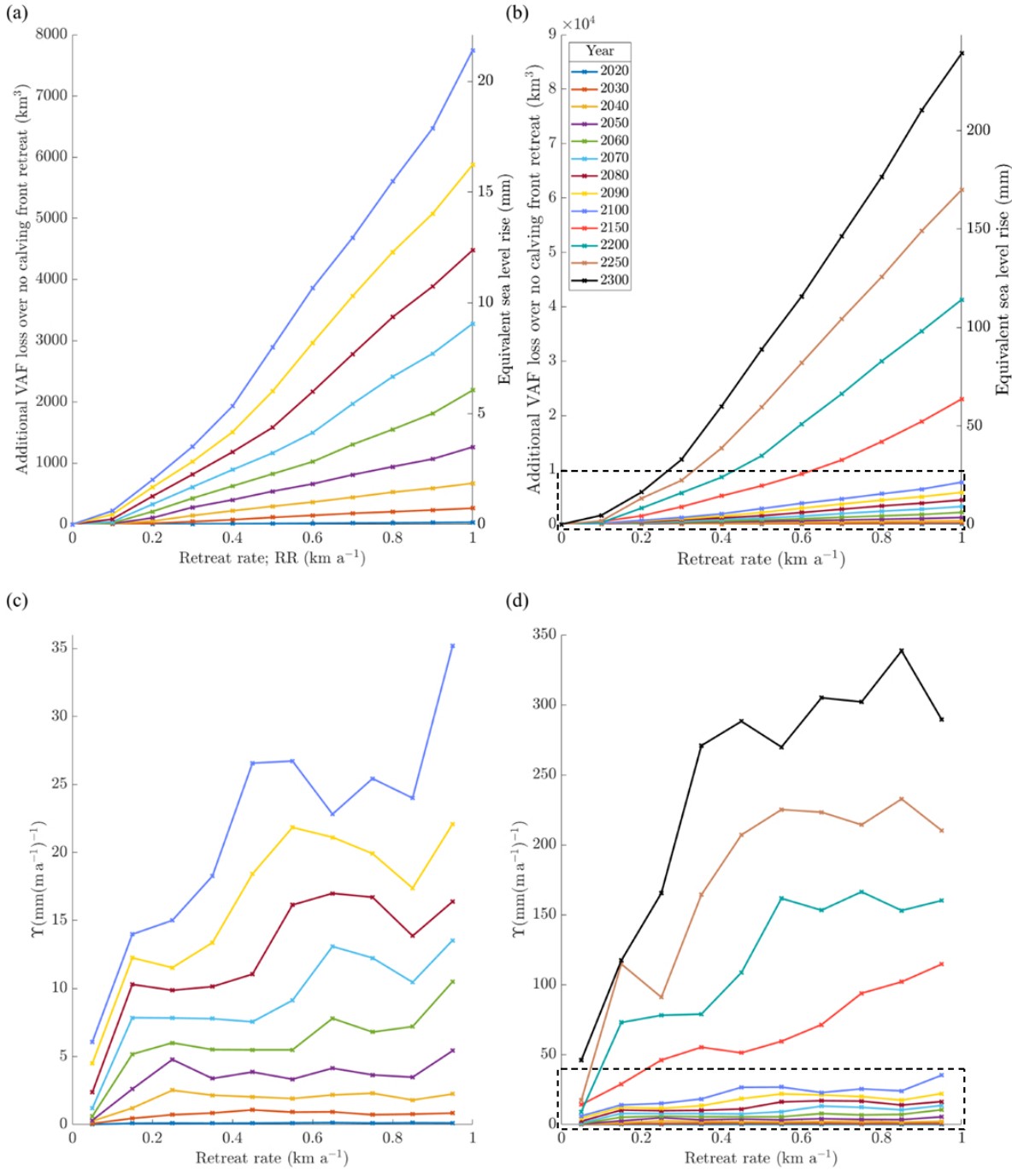

**Figure 3.** (a-b) The relationship between the retreat rate and $-\Delta\mathrm{VAF}_{\mathrm{add}}$ for the Control forcing. (c-d) The values of $\Upsilon$ calculated at each $0.1\,\mathrm{km\,a^{-1}}$ step. In all cases, each line represents a specific year during the simulations. (a,c) Years up to 2100. (b,d) Years up to 2300. The dotted outlines show the segments of panels (b,d) which are detailed in panels (a,c).

## 3.2 Responses with different retreat rates

We use -$\Delta$VAF to refer to the loss of VAF compared to the initial state in a single experiment. Introducing a prescribed retreat rate leads to -$\Delta$VAF increasing, compared to RR0. We consider the -$\Delta$VAF values of different retreat rates in comparison to that of RR0 (using the same climate forcing), in order to determine the *additional* contribution of the prescribed retreat rate. When doing so, we refer to this as $-\Delta\text{VAF}_{\text{add}}$. We give values as an equivalent contribution to mean sea level rise in millimetres (mm SLR), for ease of interpretation.

The magnitude of $-\Delta\text{VAF}_{\text{add}}$ depends on the retreat rate in a monotonic relationship, shown in Figure 3(a-b) for chosen years during the simulation, using the Control climate forcing. The relationship shows transitions between different gradients in $-\Delta\text{VAF}_{\text{add}}$, becoming steeper as the retreat rate increases in a piecewise-linear relationship. With a $1\,\text{km}\,\text{a}^{-1}$ retreat rate, $-\Delta\text{VAF}_{\text{add}}$ is $21.4\,\text{mm SLR}$ by 2100, and $239\,\text{mm SLR}$ by 2300.

Figure 3(c-d) shows the value of our *SLR-RR sensitivity*, which is proportional to the gradient of the curves in Figure 3(a-b), calculated between each $0.1\,\text{km}\,\text{a}^{-1}$ step in retreat rate. A somewhat piecewise relationship can be seen here, with visible transitions between different states of sensitivity. Such a transition occurs around $0.6\,\text{km}\,\text{a}^{-1}$ in 2060, which then occurs around $0.5\,\text{km}\,\text{a}^{-1}$ by 2080 and closer to $0.4\,\text{km}\,\text{a}^{-1}$ in 2090. Such transitions in this relationship are not always clear, but can be seen occurring at lower retreat rates over time. A second transition can be seen at around $0.9\,\text{km}\,\text{a}^{-1}$ in 2100, and then although not obvious in the curve for 2150, appears again around $0.4\text{-}0.5\,\text{km}\,\text{a}^{-1}$ in 2200.

## 3.3 Responses with different climate forcing

Figure 4(a) displays -$\Delta$VAF for each of the RR0, RR0.5 and RR1 experiments using the different climate forcings. For most of the simulation time, -$\Delta$VAF has a similar range over the different climate forcings regardless of the retreat rate. At 2100, the range is $8.08\,\text{mm SLR}$ for RR0, $8.65\,\text{mm SLR}$ for RR0.5 and $9.76\,\text{mm SLR}$ for RR1. At 2200, the ranges are $37.77\,\text{mm SLR}$, $44.25\,\text{mm SLR}$ and $46.97\,\text{mm SLR}$ respectively. By 2300, the range of RR1 has a higher value at $92.27\,\text{mm SLR}$, compared to $69.96\,\text{mm SLR}$ and $68.18\,\text{mm SLR}$ for RR0 and RR0.5, respectively. We compare these ranges – the sea level rise response to climate forcings – to the differences between responses to retreat rates as shown in Figure 4(b). We see that changing from the lowest to the highest climate forcing produces a difference in response roughly equivalent to changing from Control_RR0 to Control_RR0.5 ($7.99\,\text{mm SLR}$ at 2100, $34.75\,\text{mm SLR}$ at 2200, $88.75\,\text{mm SLR}$ at 2300), or from Control_RR0.7 to Control_RR1 ($8.46\,\text{mm SLR}$ at 2100, $47.94\,\text{mm SLR}$ at 2200, $93.02\,\text{mm SLR}$ at 2300). This shows that the full range of the response to changing our choice of climate forcing is less than the range of the response to changing our choice of calving front retreat rate.

Figure 4(b) shows $-\Delta\text{VAF}_{\text{add}}$ for all Control simulations along with the RR0.5 and RR1 experiments for all climate forcings. $-\Delta\text{VAF}_{\text{add}}$ follows a similar trajectory for each climate forcing case when using RR0.5, and the same is true for

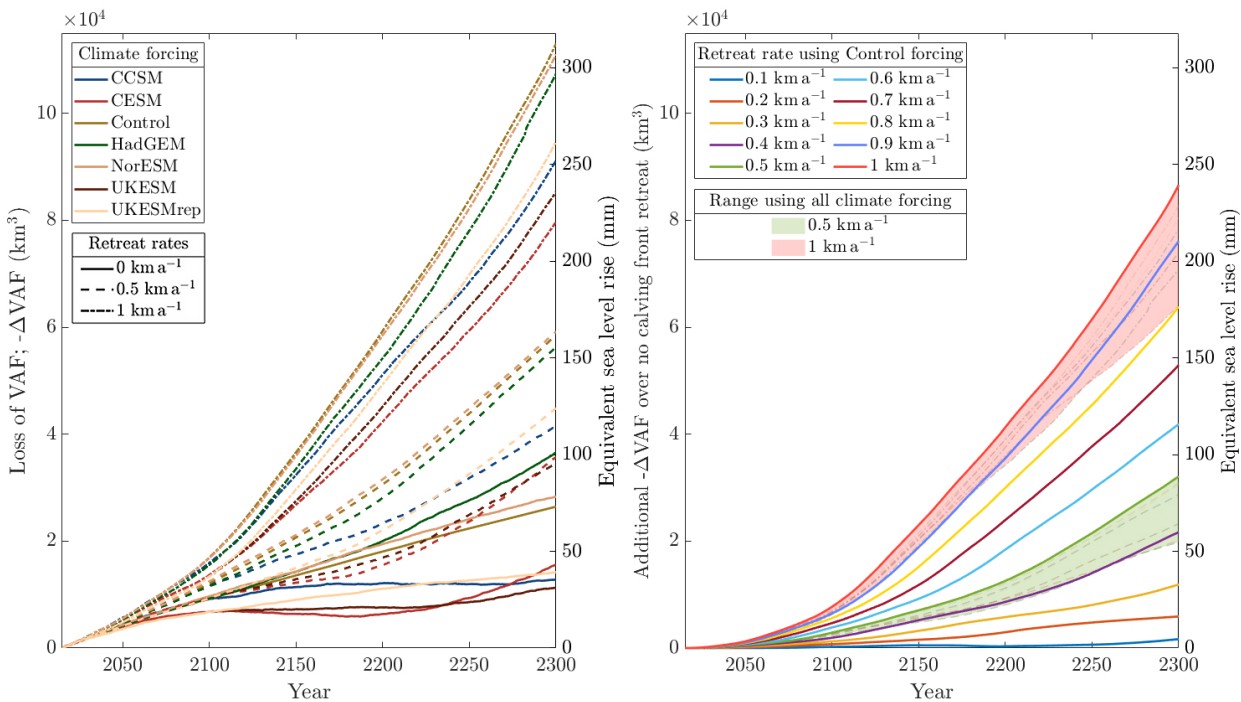

**Figure 4.** (a) $-\Delta$VAF for all RR0, RR0.5 and RR1 experiments. (b) The thick lines show $-\Delta$VAF$_\text{add}$ for all retreat rates using Control forcing, and the thin lines show $-\Delta$VAF$_\text{add}$ for all RR0.5 and RR1 experiments with shading to emphasise the full range of results using each (green for RR0.5 and red for RR1).

the RR1 simulations (as shown by the shaded areas covering the full range of these sets of simulation outputs). At 2100, for RR0.5, all climate forcings have $-\Delta$VAF$_\text{add}$ in the range 7.16 mm SLR $\pm 11.5\%$, and for RR1 the range is 19.41 $\pm 10.5\%$. At 2200, these ranges are 28.54 mm SLR $\pm 22.3\%$ and 104.37 mm SLR $\pm 9.4\%$, and at 2300 they are 71.69 mm SLR $\pm 23.8\%$ and 208.19 mm SLR $\pm 14.9\%$. So the uncertainty in the magnitude of $-\Delta$VAF$_\text{add}$ is never higher than 24%.

## 4 Discussion

### 4.1 The effects of climate forcing

Our results show that for a given prescribed non-zero retreat rate, and for the range of climate forcings in Tier 1 of ISMIP6-2300, $-\Delta$VAF$_\text{add}$ does not scale with the magnitude of -$\Delta$VAF using RR0 (i.e. in the simulations with no calving front retreat). By this we mean that regardless of the magnitude of VAF responses to climate forcing, any particular increase in the retreat rate always causes similar additional VAF loss, as demonstrated by the shaded areas in Figure 4. This demonstrates that VAF loss in models due to calving front retreat is not heavily dependent on the climate forcing. Therefore, the results displayed

in Figure 2 and Figure 3 are not entirely unique to the chosen climate forcing, which allows us to make clear statements about the impact of frontal retreat rate on the future mass loss of the region.

We suggest that this finding could potentially apply more generally to Antarctic ice shelves. Since it holds over the ASE domain, which contains a variety of ice shelf configurations from the unconstrained ice tongue of Thwaites to the highly buttressed ice shelves of Crosson and Dotson, it is possible that the choice of climate forcing, at least within the range used for ISMIP6-2300, is not strongly related to the modelled response to calving front retreat for any ice geometry. This could be the subject for a dedicated future study, and care should of course be taken in making generalisations. An obvious caveat is that in our case there is no feedback between the ice and the ocean or atmosphere. In the real world, or a coupled simulation, the rate of calving is particularly likely to impact ocean circulation and change the thermal forcing under ice shelves. Thus this finding is only relevant to stand-alone ice sheet models such as the one we use.

The climate forcings themselves produce very different results in the RR0 experiments, with some including periods of increasing VAF (Figure 4(a)). In these cases (CCSM, CESM and UKESM) the ice is thickening upstream of the grounding line, and the melt rate distribution on the ice shelf is not concentrated as close to the grounding line as it is in the cases which do not display this behaviour. It is notable that this increase in VAF does not happen before 2100 in any case, nor at all for the UKESM case with repeated forcing after 2100 (which of course does not undergo any large changes after 2100 as some of the other forcings do). This could demonstrate some limitations in the use of these extrapolated climate forcing products beyond a certain time, as grounding lines evolve and the geometry moves further away from the state used in the ocean models. This is something that could be investigated in future, but for our purposes it serves to demonstrate that even with this wide range of behaviour, $-\Delta\mathrm{VAF}_{\mathrm{add}}$ is remarkably similar between cases.

## 4.2 Thresholds in the system

The relationship between retreat rates and $-\Delta\mathrm{VAF}_{\mathrm{add}}$ exhibits a somewhat piecewise behaviour (Figure 3. The discrete steps can be attributed to the system passing certain buttressing thresholds, such as loss of contact with pinning points, which change the ice dynamics significantly enough to cause an increase in $-\Delta$VAF across the domain. In the absence of any such buttressing thresholds, the response to an increase in prescribed retreat rates remains quite linear, seen in Figure 3 as constant values of $\Upsilon$ as the retreat rate varies. This linearity is particularly obvious in earlier years of simulation, and becomes less obvious as time progresses and more complexities are introduced by the changing geometry, adding more noise to the signal in our SLR-RR sensitivity. Nevertheless, relatively flat sections can be seen, for example, from 0.1 to $0.4\,\mathrm{km\,a^{-1}}$ and again above $0.5\,\mathrm{km\,a^{-1}}$ in 2200, in which cases the value of $\Upsilon$ remains at around 75 and 150 respectively, with a transition between the two states.

There are at least two distinct thresholds which we can confidently identify in the ASE system. First, in the grounded area between Thwaites and Pine Island ice shelves, there is a peak in the bedrock geometry which is above sea level, as can be seen in Figure 2(a). As the grounding line retreats, this becomes an important pinning point which buttresses 'PIGlet'. Even-

tually, the ice loses contact with what is by that point a small island, resulting in the Pine Island and Thwaites calving fronts merging into one as Thwaites appears to be driven into retreat by 'PIGlet'. The loss of contact with this pinning point, and the resultant speed-up of Eastern Thwaites, can be seen in Figure 2(d-e). The timing of this lines up with Figure 3(c), in which a transition in the sensitivity is seen around RR0.5 in 2080. Later in the simulation, the grounding line retreat at the main trunk of PIG in Figure 2(f-g) is also instigated from the west, and may not happen without the collapse of 'PIGlet'. This threshold is never reached without calving front retreat in Control_RR0, but is reached with even the smallest prescribed retreat rate in Control_RR0.1.

A second major threshold is in the Dotson and Crosson ice shelves. They take a longer time to show major changes, but when the ice shelves lose contact with Bear Island, after about 60 km of calving front retreat, the ice flows faster and extra grounding line retreat is seen. The speed and grounding line retreat then decrease again until contact is lost with further bedrock peaks upstream of the initial grounding line. In Figure 2, the loss of contact with Bear Island and increase in speed occurs between panels (e) and (f), the following reduction in speed in panel (g) and further loss of buttressing leading to greater grounding line retreat in panel (h). The timing of this threshold appears to line up with Figure 3(c-d), in which a threshold appears in 2100 around RR0.9 and by 2200 is around RR0.4. However, the curve for 2150 does not show a clear signature to strengthen this connection. At such an advanced stage of the simulation, this could be due to several competing signals as different thresholds of varying sizes are reached. For example, several small pinning points and embayments are exposed as the grounding lines retreat, particularly around the Dotson and Crosson ice shelves, which could each be seen as a small threshold affecting buttressing.

The grounding line of the main trunk of PIG does not show any difference in position until the calving front approaches very close to it, at which point the glacier speeds up significantly. This is another threshold in the system, and occurs just after contact is lost with the Bear Island, so could be a major competing signal masking the signature in 2150 discussed above. The behaviour of Pine Island is consistent with the findings of Reese et al. (2018), Morlighem et al. (2021) and Mitcham (2022), that much of PIG's buttressing is provided by the ice closest to the grounding line.

## 4.3 Regional variability

To clarify the differing effects of individual regions of the calving front, we ran simulations with RR0.5 and RR1 in which parts of the calving front, indicated in Figure 1, were retreated individually, while keeping the rest of the calving front fixed. The results, displayed in Figure 5, show that retreat of the PIG calving front is responsible for the largest long-term changes, while retreat of the Dotson and Crosson calving fronts causes the least difference.

The effect of the entire calving front retreating is initially almost identical to the sum of the three individual experiments, but diverges as the simulations continue beyond about 2100. This roughly coincides with the point at which the ice shelves of Thwaites and Pine Island have been entirely calved away. Changes in geometry near the present-day zero-velocity boundaries

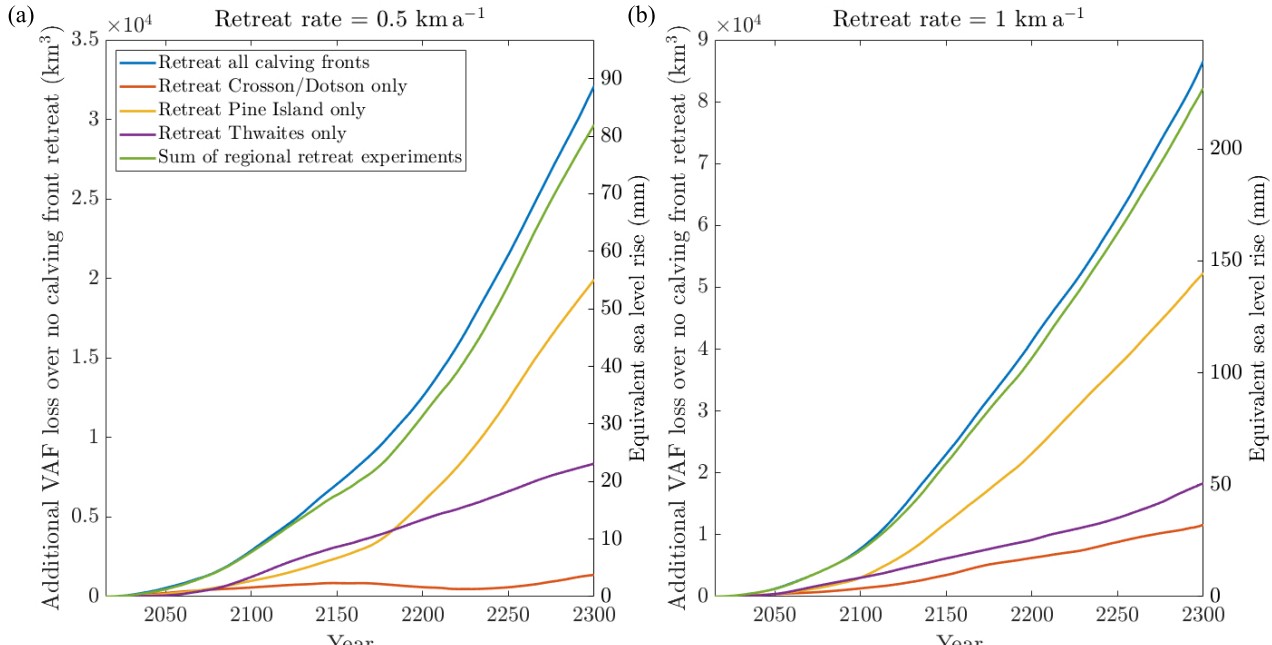

**Figure 5.** $-\Delta \mathrm{VAF_{add}}$ in experiments which only retreat defined sections of the calving front, at PIG, Thwaites or Crosson/Dotson, along with the sum of these three experiments and the result from the entire calving front moving. The sections of calving front are indicated in Figure 1

as the ice sheet evolves could be the cause of this. The present-day boundaries are used to determine where the calving front is allowed to retreat in the individual regional experiments, but these boundaries would likely not be in the exactly the same location in future, particularly around the 'PIGlet' ice flow.

270

Our results are in agreement with other recent modelling studies (e.g. Barnes and Gudmundsson, 2022; Benn et al., 2022) that the Thwaites ice shelf does not currently contribute much to the glacier dynamics. In our case, the additional VAF loss when retreating the ice front is not significant in the early stages as the ice shelf is calved away, at only 0.19 mm SLR in 2050 for RR0.5 before rising towards 2100. The changing position of the grounding line appears to be the more important factor,
275  and causes ice loss after the first few decades. Gudmundsson et al. (2023) looks specifically at the buttressing provided by Thwaites Ice Shelf, finding that fluxes across the grounding line can either increase or decrease locally when the floating ice is removed, suggesting a negligible net impact of buttressing. Naughten et al. (2023) used a buttressing flux response approach following Reese et al. (2018), finding a mixture of positive and negative responses along the Thwaites grounding line.

## 4.4 The importance, and difficulties, of calving in model predictions

With the comparisons detailed in subsection 3.3, we have demonstrated that the variability in our experimental $-\Delta$VAF outputs using retreat rates between 0 and $1\,\mathrm{km\,a^{-1}}$ is greater than that of using the range of climate forcing from the ISMIP6-2300 Tier 1 experiments. This means that accurately representing calving front movement in predictive simulations can be a problem of comparable importance to accurately representing melt rates.

There is currently no consensus on a suitable calving law to use for predictive modelling, despite several options being available as outlined in the introduction. Calving can alternatively be applied as a prescribed retreat rate, as we have done here to systematically investigate the process. However, future retreat rates are unknown so predictions using this method will be unreliable. Retreat rates assumed from historical rates can be highly variable depending on the timescales used. For example, historical calving front positions of Pine Island presented in Liu et al. (2022) show an average retreat rate over their entire record from 1973 to 2020 of $0.6\,\mathrm{km\,a^{-1}}$. But the calving front was in about the same position in 2015 as in 1973, so 5 years earlier the average retreat rate would have been almost zero. On the other hand, the most recent period of frontal retreat from 2013-2020 gives $6\,\mathrm{km\,a^{-1}}$. This observational example presents a far greater degree of uncertainty than the comparatively small range of retreat rates in our experiments.

Due to the high uncertainty in future retreat rates and the importance of this process in model evolution, we propose that explicitly testing calving mechanisms, either using existing proposed laws or prescribed retreat rates, should be a feature of future model intercomparisons. By not including such tests, a large amount of potential variability between models could be missed. More ice flow models are now able to include calving, as demonstrated by the ongoing CalvingMIP project (Jordan, 2024), so this is now a more practical option than it may have been in the past.

## 5 Conclusions

We have quantified the sensitivity of modelled VAF in the ASE to rates of imposed calving front retreat from 0.1 to $1\,\mathrm{km\,a^{-1}}$, shown in Figure 3. By 2100, using a retreat rate of $1\,\mathrm{km\,a^{-1}}$ leads to a $21.4\,\mathrm{mm}$ difference in sea level contribution compared to not including calving. For retreat rates over $0.4\,\mathrm{km\,a^{-1}}$, changing the rate by just $0.1\,\mathrm{km\,a^{-1}}$ causes around $2.5\,\mathrm{mm}$ difference in sea level contributions by 2100.

The monotonic relationship between retreat rates and the additional loss of VAF they cause is somewhat piecewise-linear, which can be explained by specific buttressing thresholds related to local geometry. We identify in particular a pinning point which appears between PIG and Thwaites as 'PIGlet' retreats, the loss of contact from which speeds up ice flow in the region and appears to instigate the collapse of Thwaites and PIG.

We have further shown that the additional loss of ice due to calving front retreat in a stand-alone ice sheet model is not heavily dependent on climate forcing (ocean-induced basal melt and surface mass balance). It can be quantified depending on the retreat rate, within bounds, as shown in Figure 4(b). For example, the additional sea level rise due to a $0.5\,\mathrm{km\,a^{-1}}$ retreat rate is $7.16\,\mathrm{mm} \pm 11.5\%$ by 2100, regardless of the chosen ISMIP6 forcing scenario.

In our experiments, the overall variability due to climate forcing is of the same order as that due to the retreat rate, showing that the two processes can be equally important considerations in predictive modelling under some circumstances. This, along with the sensitivity of modelled ice dynamics to calving front retreat rates which we have shown, highlights the importance of including calving in models. We suggest that it is important to consider variability in calving front retreat when assessing the uncertainty of predictive simulations, and we propose that use of different calving mechanisms should be explicitly included as a feature of future model intercomparison projects.

*Code and data availability.* The source code for Úa is under continuous development, and the latest version is available at https://github.com/ GHilmarG/UaSource. These experiments can be conducted using version 2023b, found at https://doi.org/10.5281/zenodo.10829346 (Gudmundsson, 2024). BedMachine v3 can be downloaded via NSIDC at https://nsidc.org/data/nsidc-0756/versions/3 (Morlighem, 2022). The ISMIP6 23rd Century Forcing Datasets can be accessed via Ghub at https://theghub.org/dataset-listing, for which an account must be created (Nowicki and ISMIP6-Team, 2024).

*Video supplement.* A supplementary video is provided showing thickness above flotation and speed for Control_RR0 (https://doi.org/10.5446 /69727). This is the run to which other Control experiments are compared in order to calculate differences. Further videos show differences in thickness above flotation and speed for Control_RR0.5 (https://doi.org/10.5446/69728) and Control_RR1 (https://doi.org/10.5446/69729), compared to Control_RR0. The grounding lines and calving fronts follow the legend of Figure 1.

## Appendix A: Modified BedMachine topography

In initial testing, we found that the grounding line of Pine Island Glacier could advance initially, which would cause it to pin on a point downstream of the present day grounding line. This is not an uncommon issue when modelling PIG, which is sensitive to uncertainties in bedrock topography (Sun et al., 2014; Wernecke et al., 2022). To ensure that the geometry in our simulations does not move quickly away from observed trends, we modified the BedMachine topography in a region under Pine Island Ice Shelf to lower the bedrock by $100\,\mathrm{m}$. This is not unreasonable, as bedrock estimates by different methods can differ by hundreds of metres (e.g. Nias et al., 2018). As we do not expect grounding line advance during our simulations, this simple uniform approach was deemed adequate, rather than smoothing the bedrock downstream of the grounding line. All elements crossing the grounding line were kept at the original BedMachine values, to avoid any change to the bedrock under grounded

340   ice. The difference between BedMachine and the bedrock elevation we use is shown in Figure A1.

There is still an initial thickening of the ice shelf, but as this no longer comes into contact with a pinning point, the grounding line does not advance from its true current position in the first years of the simulation, thus preventing an advanced grounding line position and unrealistic extra buttressing from persisting through our simulations.

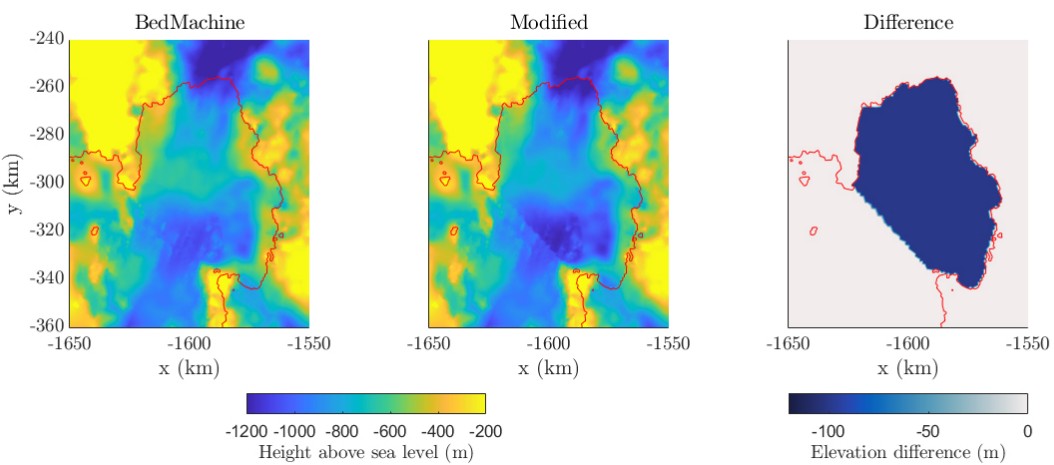

**Figure A1.** Bedrock elevation under Pine Island Ice Shelf in BedMachine and our modified geometry. The grounding line is marked in red.

## Appendix B:  Calving in Úa

345

Úa uses a level-set method to implement calving, which is summarised here. More details can be found in the Úa Compendium which is included when downloading the model (Gudmundsson, 2024).

The calving rate is a scalar quantity, defined as the difference between the retreat rate of the calving front and the velocity of
350   ice at the calving front in normal direction, $v$. We use an implicit formulation to describe the position of the calving front $\mathcal{C}$ as the solution to

$$\varphi(\mathcal{C},t) = 0 \,, \tag{B1}$$

for all times $t$, where $\varphi$ is a function $\varphi : \mathbb{R}^2 \times \mathbb{R}$. We refer to $\varphi$ as the *level-set function*. By definition, the curve moves with the (prescribed) velocity $\boldsymbol{c}$ - the calving speed or calving rate - in a direction $\hat{\boldsymbol{n}}$ normal to the curve $\mathcal{C}$,

355   $$\boldsymbol{c} = c\hat{\boldsymbol{n}} \tag{B2}$$

$$= -c\frac{\nabla\varphi}{\|\nabla\varphi\|} \,, \tag{B3}$$

where the normal vector is

$$\hat{\boldsymbol{n}} = -\frac{\nabla\varphi}{\|\nabla\varphi\|} \,, \tag{B4}$$

with

$$\|\nabla\varphi\|^2 = \nabla\varphi \cdot \nabla\varphi \,. \tag{B5}$$

The sign convention used in the definition of the normal in Equation B4 is introduced in the anticipation that $\varphi$ will be defined as a decreasing function of distance as we travel across the calving front, from the ice covered region to the ice-free region, with the normal $\hat{\boldsymbol{n}}$ pointing outwards. Using this sign convention we find that

$$\boldsymbol{c} \cdot \nabla\varphi = c\,\hat{\boldsymbol{n}} \cdot \nabla\varphi \tag{B6}$$

$$= -c\|\nabla\varphi\| \,. \tag{B7}$$

The velocity, $\boldsymbol{u}$, of the calving front $\mathcal{C}$ is equal to the difference between the material velocity, $\boldsymbol{v}$, of ice at the calving front and the calving velocity $\boldsymbol{c}$, that is

$$\boldsymbol{u} = \boldsymbol{v} - \boldsymbol{c} \,. \tag{B8}$$

As $\varphi$ must not change for any point along the curve travelling with the velocity $\boldsymbol{u}$,

$$\varphi(\boldsymbol{u}(\boldsymbol{x},t),t) = K \,, \tag{B9}$$

where $K$ is some constant independent of $t$, and therefore

$$\partial_t\varphi + (\boldsymbol{v} - \boldsymbol{c}) \cdot \nabla\varphi = 0 \,. \tag{B10}$$

Rearranging and using Equation B7, this can be written as

$$\partial_t\varphi + \boldsymbol{v} \cdot \nabla\varphi = -c\|\nabla\varphi\| \,. \tag{B11}$$

Equation B10 and Equation B11 are different forms of the kinematic calving front condition. When used to calculate the zero level of $\varphi$, we refer to it as the *level-set equation*.

A level set method based on a variational principle can be derived by adding a perturbation to the energy potential (Luo et al., 2019). Minimising this additional potential term involves adding the corresponding directional derivative with respect to $\varphi$ to Equation B11, resulting in the *augmented level set-equation*

$$\partial_t \varphi + \boldsymbol{v} \cdot \nabla \varphi - \nabla \cdot (\kappa \nabla \varphi) = -c\|\nabla \varphi\|\,, \tag{B12}$$

where $\kappa$ is a diffusion coefficient. In our case $\kappa$ takes the form

$$\kappa = \mu k(\nabla \varphi), \tag{B13}$$

where

$$k(\nabla \varphi) = \|\nabla \varphi\|^2 - 1, \tag{B14}$$

with $\mu = 0.2$.

In Úa the level set is evolved at every time step by solving the augmented level-set equation implicitly with respect to $\varphi$ using the Newton-Raphson method (NR) with consistent Streamline Upwind Petrov-Galerkin (SUPG) weighting. This has been shown to be an effective stabilisation method (Cheng et al., 2024).

For a migrating calving front the ice downstream needs to be calved away. This is done using a melt rate parameterisation in which an additional melt rate is prescribed implicitly as a function of the ice thickness. We use

$$a_c = (1 - \mathcal{H}(\varphi))\, a_1 (h - h_{\min})\,, \tag{B15}$$

where $a_c$ is the additional melt rate, $\mathcal{H}$ is the Heaviside step function and $h_{\min}$ is the desired minimum ice thickness. $a_1$ is a constant such that ice is removed within the time $1/|a_1|$, for $a_1 < 0$. We set $a_1 = -10$.

*Author contributions.* JMB and GHG designed the experiments, which were implemented and run in the model by JMB. SS prepared the ISMIP6 forcing datasets. JMB, GHG and DNG contributed to interpretation and discussion of results. JMB wrote and prepared the manuscript, with contributions from all authors.

*Competing interests.* At least one of the (co-)authors is a member of the editorial board of The Cryosphere. The peer-review process was guided by an independent editor, and the authors have no other competing interests to declare.

*Acknowledgements.* This work is from the PROPHET project, a component of the International Thwaites Glacier Collaboration (ITGC). Support from Natural Environment Research Council (NERC: Grants NE/S006745/1, NE/S006796/1). This is ITGC Contribution No. ITGC-128. Additional support has been received from Novo Nordisk Foundation grant NNF23OC0081251. The authors would like to thank Kerim

410    Nisancioglu for handling the editing of this paper, two anonymous referees for their constructive reviews, and Matt Trevers for his very useful

comments on a previous iteration of the manuscript.

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
