# Peer review of "Modelling the sensitivity of ice loss to calving front retreat rates in the Amundsen Sea Embayment, West Antarctica"

_EGUsphere, 2025_

## Referee Comment (RC1)

**Summary:**

This manuscript investigates the sensitivity of ice loss in the Amundsen Sea Embayment (ASE), West Antarctica, to prescribed calving front retreat rates using the Úa ice-flow model. The authors systematically apply constant retreat rates from 0.1 to 1 km/year along the calving fronts of major glaciers, including Pine Island, Thwaites and Crosson/Dotson. They quantify the additional sea-level rise (SLR) contributions caused by these retreat rates, comparing their impact with that of ocean-induced melt variability from ISMIP6 ocean forcing scenarios. They find that calving processes have an impact comparable to ocean forcing, highlighting the necessity of better representation of calving in future ice sheet models.

This manuscript addresses a crucial gap in understanding the dynamics of Antarctic ice shelves, specifically the sensitivity of future ice loss projections to calving front retreat. The study is timely and of high relevance, given the uncertainties associated with calving dynamics and their impact on global sea-level rise. However, parts of the results and discussion lacks clarity. This overall issue is address in more detail in the specific comments below. Several aspects would benefit from additional clarifications and expansions to enhance the robustness and readability of the paper.

**Specific comments:**

L17-18: Please add references.

L21: Does this mean that it is easier to implement continuous calving rate rather than modeling small-scale individual calving events? Please clarify.

L31: "…reproducing *observed grounding lines* consistently…" => reproducing observed calving front

L32: More numerical models now include calving capability, as mentioned in the Discussion section. Therefore, it may be more appropriate to specify "for real glacier simulations?". Also please add some supporting references.

L47: "raise sea levels by over a metre." : Please add references.

L62-69: This should move to the Methods section.

L67: What ocean domain is referred to here? I assume it means the surface area of the entire global ocean — please clarify.

L76: Why was it necessary to lower the bed topography when forcing the retreat? By how much was it lowered? How does the model behavior differ if this step is included?

L107: "The repeated forcing… between 2100 and 2300.": Could you clarify this? Was the forcing from 2080–2100 repeated uniformly across the 2100–2300 period?

L103-108: It would be helpful to include a table summarizing the experiments described here or in Section 2.3. For the ocean forcing scenarios, please clarify which models provide extension to 2300 and which provide data only for a partial period and should be extended with repeated forcing.

L114: 'Ocean_RR#' this format is not used later in the text, which make it difficult to follow the experiment descriptions in the Results and Discussion sections.

L128: "… at various points during the simulation": … at various points ***in time*** during the simulation.

L141: a significantly different ***compared to Control_RR0***?

L142: until later: Please specify the year or time range.

L146: "as contact is lost with pinning points": Please specify the year after which contact is lost with pinning points.

L170: … SLR for RR0 and RR0.5***, respectively***.

L170: "Comparing these ranges to Figure 3(b)…" Consider rewriting this sentence to clarify what is being compared here. What's the range of RR0 to RR0.5 or RR0.7 to RR1?

L173-178: This paragraph is unclear. For example, "-ΔVAF_add follows a similar trajectory for each RR00.5 and each RR1 ocean forcing case." Does this mean "-ΔVAF_add of Control_RR1 falls within the upper bound of the RR0.5 ensemble across ocean forcings"?  Please rephrase for clarity.

L181: "Our results show that for a given … without retreat.": Could you give a figure or table including -ΔVAF  and -ΔVAF_add for each ocean forcing scenario to strengthen this statement?

L186: I think this sentence makes too strong a generalization. Perhaps, I suggest something like "While our findings demonstrate that prescribed calving front retreat rates drive comparable ice loss across a range of ISMIP6-2300 ocean forcings within the ASE, caution should be taken in generalizing these results to the entire Antarctic Ice Sheet."

L204: Refer to Figure 2(c) and (d)

L229: RR0.4

L229: "However, the curve for..": can you comment more on this?

L229-L240: Consider moving this part to the Results section.

L241: Can you comment more on "some interactions"?

L246-L247 "In our case, the additional VAF loss when retreating…" Please quantify the additional VAF until 2050 or 2100 (early stages).

L255: Could the post 2100 results be influenced by the repeated ocean forcing scenarios? If so, it may be worth discussing this.

---

## Author Comment (AC1)

Response to RC1

Author responses are in blue.

Summary:

This manuscript investigates the sensitivity of ice loss in the Amundsen Sea Embayment (ASE), West Antarctica, to prescribed calving front retreat rates using the Úa ice-flow model. The authors systematically apply constant retreat rates from 0.1 to 1 km/year along the calving fronts of major glaciers, including Pine Island, Thwaites and Crosson/Dotson. They quantify the additional sea-level rise (SLR) contributions caused by these retreat rates, comparing their impact with that of ocean-induced melt variability from ISMIP6 ocean forcing scenarios. They find that calving processes have an impact comparable to ocean forcing, highlighting the necessity of better representation of calving in future ice sheet models.

This manuscript addresses a crucial gap in understanding the dynamics of Antarctic ice shelves, specifically the sensitivity of future ice loss projections to calving front retreat. The study is timely and of high relevance, given the uncertainties associated with calving dynamics and their impact on global sea-level rise. However, parts of the results and discussion lacks clarity. This overall issue is address in more detail in the specific comments below. Several aspects would benefit from additional clarifications and expansions to enhance the robustness and readability of the paper.

Thank you for your review, and your positive comments on the study along with constructive criticism on the clarity of presentation. In the revised manuscript, efforts have been made to improve clarity in all instances you have pointed out, in order to make the results and discussion easier for a reader to follow and present our work in the best possible way.

Specific comments:

L17-18: Please add references.

References added to Depoorter et al. (2013) and Pattyn et al., (2017).

L21: Does this mean that it is easier to implement continuous calving rate rather than modelling small-scale individual calving events? Please clarify.

We have reworded this to "it is often easier and more practical to implement calving in long-term simulations via a continuous calving rate, rather than modelling individual calving events as they occur naturally."

L31: "...reproducing observed grounding lines consistently..." => reproducing observed calving front

Fixed. Thanks!

L32: More numerical models now include calving capability, as mentioned in the Discussion section. Therefore, it may be more appropriate to specify "for real glacier simulations?". Also please add some supporting references.

Updated to specify "for predictive simulations", and added a reference to ISMIP6 as an example: "For example, in the recent model intercomparison project ISMIP6 (Seroussi et al., 2020), only three of the ten participating models implement a more complex calving scheme. One applies an approach based on strain rates (Levermann et al., 2012), while two other models use the calving law of Pollard et al. (2015)."

L47: "raise sea levels by over a metre." : Please add references.

Reference added: Rignot et al. (2002).

L62-69: This should move to the Methods section.

This has been moved as suggested.

L67: What ocean domain is referred to here? I assume it means the surface area of the entire global ocean — please clarify.

We have clarified this is global.

L76: Why was it necessary to lower the bed topography when forcing the retreat? By how much was it lowered? How does the model behavior differ if this step is included?

Information on this, plus a figure showing the difference between out topography and BedMachine, has been added to a new appendix.

L107: "The repeated forcing... between 2100 and 2300.": Could you clarify this? Was the forcing from 2080–2100 repeated uniformly across the 2100–2300 period?

This has been clarified: "...sampled randomly between 2100 and 2300 to avoid repeating the same forcing pattern."

L103-108: It would be helpful to include a table summarizing the experiments described here or in Section 2.3. For the ocean forcing scenarios, please clarify which models provide extension to 2300 and which provide data only for a partial period and should be extended with repeated forcing.

A table has been included to summarise this.

L114: 'Ocean_RR#' this format is not used later in the text, which make it difficult to follow the experiment descriptions in the Results and Discussion sections.

This was an oversight, and we have edited throughout to stick to this format where possible.

L128: "… at various points during the simulation": … at various points in time during the simulation.

Added

L141: a significantly different compared to Control_RR0?

Clarified

L142: until later: Please specify the year or time range.

Added

L146: "as contact is lost with pinning points": Please specify the year after which contact is lost with pinning points.

This whole section has been improved by adding references to panels in the figure as well as specifying years.

L170: … SLR for RR0 and RR0.5, respectively.

Added

L170: "Comparing these ranges to Figure 3(b)…" Consider rewriting this sentence to clarify what is being compared here. What's the range of RR0 to RR0.5 or RR0.7 to RR1?

This has been reworded, and additional values added in for the ranges.

L173-178: This paragraph is unclear. For example, "-DVAF_add follows a similar trajectory for each RR00.5 and each RR1 ocean forcing case." Does this mean "-DVAF_add of Control_RR1 falls within the upper bound of the RR0.5 ensemble across ocean forcings"? Please rephrase for clarity.

This has been reworded, and now references the relevant part of the figure as a visual aid to clarify the point. We have also changed the caption on the figure for clarity.

L181: "Our results show that for a given … without retreat.": Could you give a figure or table including -DVAF and -DVAF_add for each ocean forcing scenario to strengthen this statement?

We have added a clarification of the statement with reference to the figure to demonstrate our meaning.

L186: I think this sentence makes too strong a generalization. Perhaps, I suggest something like "While our findings demonstrate that prescribed calving front retreat rates drive comparable ice loss across a range of ISMIP6-2300 ocean forcings within the ASE, caution should be taken in generalizing these results to the entire Antarctic Ice Sheet."

We believe the potential of generalisation is an important possibility to point out, so have kept this in. We suggest it as an interesting subject for future work, rather than asserting it as fact. Further to the existing caveats in the paragraph, an extra clause has been added that care should of course be taken in making generalisations.

L204: Refer to Figure 2(c) and (d)

Added

L229: RR0.4

Corrected

L229: "However, the curve for..": can you comment more on this?

An additional comment has been added, suggesting that competing signals from various sources could mask the particular threshold signal we are looking for.

L229-L240: Consider moving this part to the Results section.

We have left this in place, but updated the paragraph to strengthen its link to the rest of the discussion in this section.

L241: Can you comment more on "some interactions"?

We have rewritten this sentence to highlight that the sum of the three individual experiments does not differ from our main experiments until around ~2100, and speculate that it is due to our boundary definitions.

L246-L247 "In our case, the additional VAF loss when retreating..." Please quantify the additional VAF until 2050 or 2100 (early stages).

This has been added for 2050.

L255: Could the post 2100 results be influenced by the repeated ocean forcing scenarios? If so, it may be worth discussing this.

The UKESM repeated forcing certainly produces different results to the non-repeating forcing, as can be seen in Fig.3(a). However, this is only one of the range of forcings we use, and this particular difference is not important for our results beyond being part of demonstrating that the retreat rates are largely independent of the chosen forcing.

References

Depoorter, M. A., Bamber, J. L., Griggs, J. A., Lenaerts, J. T., Ligtenberg, S. R., van den Broeke, M. R., and Moholdt, G.: Calving fluxes and basal melt rates of Antarctic ice shelves, Nature, 502, 89–92, 2013.

Levermann, A., Albrecht, T., Winkelmann, R., Martin, M., Haseloff, M., and Joughin, I.: Kinematic first-order calving law implies potential for abrupt ice-shelf retreat, The Cryosphere, 6, 273–286, 2012.

Pattyn, F., Favier, L., Sun, S., and Durand, G.: Progress in numerical modeling of Antarctic ice-sheet dynamics, Current climate change reports, 3, 174–184, 2017.

Pollard, D., DeConto, R. M., and Alley, R. B.: Potential Antarctic Ice Sheet retreat driven by hydrofracturing and ice cliff failure, Earth and Planetary Science Letters, 412, 112–121, 2015.

Rignot, E., Vaughan, D. G., Schmeltz, M., Dupont, T., and MacAyeal, D.: Acceleration of Pine island and Thwaites glaciers, west Antarctica, Annals of Glaciology, 34, 189–194, 2002.

Seroussi, H., Nowicki, S., Payne, A. J., Goelzer, H., Lipscomb, W. H., Abe-Ouchi, A., Agosta, C., Albrecht, T., Asay-Davis, X., Barthel, A., et al.: ISMIP6 Antarctica: a multi-model ensemble of the Antarctic ice sheet evolution over the 21st century, The Cryosphere, 14, 3033–3070, 2020.

---

## Author Comment (AC2)

Author responses are in blue.

In this study, the authors address a simple yet important question: What is the sensitivity of ice loss to constant, uniform calving front retreat rates in the Amundsen Sea Embayment? Calving is a major source of uncertainty in ice-sheet model projections due to the lack of robust and consistent calving laws. However, this question had not been addressed yet. By filling this gap and helping to assess the influence of calving on the loss of ice shelf buttressing capacity and, consequently, on ice sheet ice loss, this study makes a highly relevant and valuable contribution to the community.

One of the strengths of the study is the 'simplicity' of its experimental design. By prescribing constant retreat rates using the ice sheet model Ua, the authors are able to identify key features of the ice sheet response, such as specific buttressing thresholds, and demonstrate how the magnitude of the retreat modulates mass loss. The paper is very well written, concise, and easy to follow. The authors have a clear objective, which is clearly stated from the beginning, and address it in a consistent and coherent way.

Thank you for your review, and your positive comments on the study. I'm glad that the way we have approached this is appreciated, and that it will be seen as a valuable contribution.

My only concern relates to one of the study's key messages. By comparing simulations forced by ISMIP6-2300 Tier 1 ocean/atmosphere coupled model outputs with the range obtained by varying the prescribed retreat rate, the authors conclude that accurately representing calving front retreat may be as important as accurately representing sub-shelf melt rates. However, if I understand correctly, the simulations are forced using both atmospheric and oceanic outputs following the ISMIP6-2300 protocol. Therefore, the spread in simulations forced by different ESMs cannot be attributed only to ocean forcing, as it also reflects changes in surface mass balance. Typically, by 2300 under high emission scenarios, some ESMs project significant increases in both snow accumulation and surface runoff. Moreover, by accounting for a range of ESM forcing, it seems to me that the authors are addressing the variability in climate forcings themselves, rather than isolating the influence of the way ocean-induced melt is accounted for in ice-sheet models. Either the message should be reformulated to something like 'the variability in the range of ice sheet response using different retreat rates is similar to the spread resulting from a range of ESM forcings', or the methodology applied to compare the impact of retreat rate and ocean-induced variability should be adjusted. This could be done, for example, by comparing to the spread in ice sheet response obtained using different sub-shelf melt parameterisations, or varying the

gamma0 value within a given parameterisation (such as the local quadratic one used here).

This is a very good point. Thank you for raising the issue. The simulations are indeed forced by both ocean and atmospheric changes from the ESMs, and in our analysis of results we have erroneously attributed all of the variability to ocean forcing. In the revised manuscript, we will be careful to point out at the first opportunity that the spread is due to both ocean and SMB forcing, and alter the wording of all following statements to refer to "climate forcing" instead of just "ocean forcing". It is also true that we are not addressing the issue of how ocean-induced melt is introduced to models, but looking at a range of different forcings which are introduced in the same way. We did not mean to give the impression that we were aiming to do the former, and will rephrase where necessary to make our intentions and the appropriate interpretation of our results clearer.

Once this point is addressed, along with the minor comments listed below, I believe this study will make a very valuable contribution to the community.

**Specific comments**

l.10-12: As explained above, I find the comparison between variability attributed to retreat rates and the one attributed to ocean-induced melt misleading.

Agreed. This sentence is rewritten as "We compare the variability in our results using different retreat rates to that when using ocean melt and surface mass balance forcing derived from different earth system models for ISMIP6, as these forcings are major factors in determining the future evolution of the Antarctic ice sheet."

l.76-77: It would be helpful to provide more details on this adjustment of the Bedmachine dataset, perhaps in the supplementary materials.

Yes, this will be included in an appendix in the revised version and referred to here.

l.110-115: Consider adding a table summarising the experiments, for clarity.

We have added a table into the revised manuscript.

Figure 1: Since Figures 1a–b and 1c–i convey different messages, you might consider splitting them into two separate figures: one introducing the study area and another focused on the results.

This is a fair point. We will rearrange the figures to avoid any confusion arising from multiple messages being in the same one.

Figure 2: You could try making the figure more self-explanatory, for example, by highlighting in panels (b) and (d) the regions zoomed in by panels (a) and (c).

A good idea. We will do this.

l.165: You refer to the ESM forcings as 'ocean forcing', but I believe that you also include atmospheric forcing through anomalies in SMB following the ISMIP6 protocol? If so, the term 'ocean forcing' may be misleading, as changes in SMB also have quite a significant influence on the ice sheet response.

This is being addressed as per previous comments.

l.181-182: Do you mean here that the ~70 mm range for RR0 in Figure 3a is larger than the ranges shown by the green and red shaded areas in Figure 3b? If so, it would be helpful to specify that explicitly for clarity.

This has been clarified with an additional sentence with a reference to the figure.

l.198-201: Could this be explained by increases in SMB post-2100 in UKESM as compared to UKESMrep?

A quick look at the mass balance shows a large difference in basal melt between the two post-2100, and greater variability in SMB though not a large difference in magnitude. We've added a clarification that of course the UKESMrep forcing does not undergo any large changes after 2100, as opposed to other models.

l.209: replace 'sensitivity' by 'SLR-RR-sensitivity' for clarity.

Done

l.220-221: I'm not sure where to visualise that the threshold is reached even for the smallest prescribed retreat rate. Clarification or reference to a figure would be useful.

We've clarified to specify that we're talking about the RR0.1 simulations. We decided for simplicity to focus mainly on RR0.5 and RR1 in our figures and discussion, so this specific simulation is not on a figure in the paper. We will make the model outputs available for anybody interested in looking at them in more detail.

l.255-258: In line with my previous comments, I think that this statement should be revised to something like 'greater than the range of forcing from the ISMIP6-2300 Tier 1 experiments'. It does not seem to me that the current experiments allow for direct comparison between the influence of calving front retreat variability and variability in sub-shelf melt representation in ice sheet models. Instead, you account for variability in climate forcings, including ocean and atmospheric forcings.

l.286-296: Same as above.

This has been altered in both cases.

---

## Referee Report (RR1)

**Summary:**

The revised version of manuscript clearly addresses the reviewers' comments raised during the first round of review. It improves the clarity of both results and discussion. Below are some minor issues that I believe could further improve the manuscript.

**Specific comments:**

L178: What range is being referred to, specifically? I assume it is the range of SLR values for RR1?

L178-181: Why are these two ranges being compared? What is the purpose or significance of this comparison? Currently, I don't see further discussion on this point. I believe expanding on this comparison would improve clarity and help strengthen the interpretation of the results.

L245: Could you add more details about the curve for the 2150? What competing signals are affecting its behavior?

L261: Does 'changes in geometry' refer to the lowering of bed topography as explained in Appendix A? Please clarify.

---

## Author Response (AR2)

**Response to reviewer comments**

**(Author responses in blue)**

**Summary:**

The revised version of manuscript clearly addresses the reviewers' comments raised during the first round of review. It improves the clarity of both results and discussion. Below are some minor issues that I believe could further improve the manuscript.

**Specific comments:**

L178: What range is being referred to, specifically? I assume it is the range of SLR values for RR1?

We have clarified the sentence here to explicitly state that we are comparing the range of responses to climate forcing and the range of responses to calving front retreat rate.

L178-181: Why are these two ranges being compared? What is the purpose or significance of this comparison? Currently, I don't see further discussion on this point. I believe expanding on this comparison would improve clarity and help strengthen the interpretation of the results.

This acts as a comparison between the full range of the response to changing our climate forcing choice, and the range of response to changing the calving front retreat rate. This point is picked up in section 4.4, but it was not obvious so the text has been updated to explicitly state the connection.

L245: Could you add more details about the curve for the 2150? What competing signals are affecting its behavior?

An example has been added of potential causes for competing signals, specifically numerous small pinning points and embayments which would affect buttressing as the grounding line retreats.

L261: Does 'changes in geometry' refer to the lowering of bed topography as explained in Appendix A? Please clarify.

This refers to changes in the location of the zero-velocity boundaries between basins over time, and the text has been edited to clarify the point.